# MRI-Based Deep Learning Tools for MGMT Promoter Methylation Detection: A Thorough Evaluation

**DOI:** 10.3390/cancers15082253

**Published:** 2023-04-12

**Authors:** Lucas Robinet, Aurore Siegfried, Margaux Roques, Ahmad Berjaoui, Elizabeth Cohen-Jonathan Moyal

**Affiliations:** 1IRT Saint-Exupéry, 31400 Toulouse, France; 2IUCT-Oncopole-Institut Claudius Regaud, 31100 Toulouse, France; 3INSERM UMR 1037, Cancer Research Center of Toulouse (CRCT), University Paul Sabatier Toulouse III, 31100 Toulouse, France; 4Pathology and Cytology Department, CHU Toulouse, IUCT Oncopole, 31100 Toulouse, France; 5Department of Neuroradiology, Hopital Pierre Paul Riquet, CHU Purpan, 31300 Toulouse, France

**Keywords:** glioblastoma, MGMT promoter, deep learning, confidence

## Abstract

**Simple Summary:**

A major prognosis factor for glioblastoma patients is the methylation status of the DNA repair enzyme MGMT. Obtaining this information using deep learning models trained on non-invasive MRI data is a major challenge with no scientific consensus to date. In this study, we provide a more rigorous and comprehensive answer to this question by using confidence metrics and relating them to the exact percentage of methylation obtained at biopsy. This systematic approach confirms that the deep learning algorithms developed until now are not suitable for clinical application. We also provide, to the best of our knowledge, the first fully reproducible source code and experiments on this issue.

**Abstract:**

Glioblastoma is the most aggressive primary brain tumor, which almost systematically relapses despite surgery (when possible) followed by radio-chemotherapy temozolomide-based treatment. Upon relapse, one option for treatment is another chemotherapy, lomustine. The efficacy of these chemotherapy regimens depends on the methylation of a specific gene promoter known as MGMT, which is the main prognosis factor for glioblastoma. Knowing this biomarker is a key issue for the clinician to personalize and adapt treatment to the patient at primary diagnosis for elderly patients, in particular, and also upon relapse. The association between MRI-derived information and the prediction of MGMT promoter status has been discussed in many studies, and some, more recently, have proposed the use of deep learning algorithms on multimodal scans to extract this information, but they have failed to reach a consensus. Therefore, in this work, beyond the classical performance figures usually displayed, we seek to compute confidence scores to see if a clinical application of such methods can be seriously considered. The systematic approach carried out, using different input configurations and algorithms as well as the exact methylation percentage, led to the following conclusion: current deep learning methods are unable to determine MGMT promoter methylation from MRI data.

## 1. Introduction

Glioblastoma represents one of the most common and most agressive malignant primary brain tumor [1]. Despite surgery (when possible), followed by the combination of radiotherapy and chemotherapy [2], these tumors remain fatal because of a systematic local relapse due to resistance to treatments. One major prognosis factor is the DNA repair enzyme O(6)-methylguanine-DNA methyltransferase (MGMT) methylation status [3]. Indeed, MGMT reduces the efficiency of alkylating chemotherapies as temozolomide and lomustine, which for temozolomide when coupled with surgical resection and radiotherapy, represents the first line treatment [1] and for lomustine is a standard treatment at relapse. The MGMT promoter methylation plays a major role in the level of gene expression and, de facto, impacts the clinical decision [4]. Determining MGMT methylation status is currently performed on the tumor when possible or on a biopsy. Tumor heterogenity and low material quantity can render MGMT status analysis difficult (even non-representative), when working only on a biopsy. Moreover, in case of relapse, a second resection is not regularly performed and the MGMT status is usually not known on the recurrent tumor. Predicting this biomarker from a non-invasive technique such as the analysis of anatomical MRIs, routinely used for evaluation of central nervous tumors according to the international RANO criteria [5], would represent an important step forward for personalised medicine. Some studies show a clear association between visually assessed ring enhancement and unmethylated patients [6,7]. Drabycz et al. [6] also determined significant differences between methylated and unmethylated patients on texture features derived from T2w scans. Moreover, when considering MR features obtained by two neuroradiologists to predict the MGMT promoter status with statistical techniques such as Fisher’s exact test and logistic regression, Han et al. [8] find out that MGMT promoter methylation was associated with tumor location and necrosis. However, Mikkelsen et al. [9] suggests significant associations between the promoter MGMT status and images features (necrotic volumes, percentage of necrosis, tumor volumes..) derived from T1wCE imagery, not allowing for scientific consensus. Finally, the contrasting results obtained by deep learning algorithms estimating MGMT promoter status from voxel intensities or feature radiomics fuel this blurring and suggest more robust and in-depth research. In this ambiguous context, the use of robust and trusted methods make a difference. Indeed, they can explain the discrepancies in results found in the literature by determining the samples on which the model performs very well and the others. This would allow us to see if the prediction of MGMT promoter status is easier to find on some patients than others, thus explaining why the results are impressive on some analyses [10,11,12,13,14] and disappointing on others [15,16]. Furthermore, in a longer term vision, in an ambiguous context such as this, providing the confidence associated with each prediction of the model is vital to potentially assist the clinician in their diagnosis.

## 2. Related Work

### 2.1. MGMT Promoter Methylation

With all the issues surrounding the classification of the MGMT promoter gene methylation, extensive efforts have been made to extract information from MRI scans, yielding promising results in some studies. For instance, Chang et al. [12] use regular residual networks [17] based on T2-weighted (T2w), FLAIR, T1-weighted (T1w), and T1-weighted with enhanced contrast (T1wCE) and achieve 83% accuracy on a five-fold cross-validation set. Moreover, Yogananda et al. [10] report a mean cross-validation accuracy of nearly 95%, which supports the previous findings. The databases used in these studies are from The Cancer Imaging Archives (TCIA) [18] for MR imaging and The Cancer Genome Atlas (TCGA) [19] for genomic information. On another dataset, a ResNet50 demonstrates 94.90% accuracy [11]. In addition, other techniques based on radiomics, the large-scale analysis of medical imaging data, have emerged to predict MGMT promoter methylation. Indeed, by working with the 704 radiomic features of 53 patients extracted from multimodal MRI scans, Le et al. [14] achieve 89% accuracy with a solution based on the F-scores for dimensionality reduction and XGBoost for classification. The work of Do et al. [13] is based on a similar method but uses a genetic algorithm to optimize the selection of radiomic features. This trick makes it possible to raise the score to an accuracy close to 93%.

However, the study conducted by Saeed et al. [15] is much more pessimistic. Indeed, several state-of-the-art deep learning models stumble upon this classification task. The best and only results are around a 0.60 AUC score, which is only slightly better than a random classifier. Moreover, results from the BraTS21 competition [16], hosted by the Radiological Society of North America (RSNA) and the Medical Image Computing and Computer-Assisted Interventions (MICCAI), reinforce these fears. Despite having one of the largest datasets, the first place was achieved with modest AUC score of 0.62, while other participants closely trailed behind with values that were slightly better than random.

### 2.2. Confidence Scores & Out-of-Distribution (OOD) Detection

Deep learning models tend to predict very high and completely inconsistent output values. Hendrycks et al. [20] demonstrate that a MNIST-based classifier can output a 0.91 softmax prediction when fed with gaussian noise. This behavior is not acceptable in a critical context such as clinical practice. Confidence metrics have emerged to deal with this issue. A scalar c∈[0,1] is assigned to each prediction reflecting the confidence that the model places in its outputs. We denote T the training set, c:T→[0,1] the confidence metric and $C_{t}=\left\{x\in T|c\left(x\right)>t\right\}$ the subset where each sample has a confidence score above an arbitrary threshold *t*. Let $m_{t}$ denote a usual metric such as accuracy, precision or recall on $C_{t}$. We want a confidence metric $M:t\mapsto m_{t}$ to be monotonically increasing. Out-of-distribution (OOD) detection and confidence areas are deeply related since the confidence score must keep the same behavior and assign low scores to out-of-distribution samples. Hendrycks et al. [20] demonstrate that a deep classifier has a lower maximum softmax probability on anomalous samples or unconfident predictions. A better alternative, introduced by Liang et al. [21], consists in calibrating the softmax layer with a temperature and process every input *x* to push it closer to the predicted class $\tilde{x}=x-\epsilon sign(-\nabla_{x}log\left(S_{\hat{y}}\left(x;T\right)\right)$ where $S_{\hat{y}}\left(x;T\right)$ is the output of the softmax layer with temperature for the input *x*. This method has the advantage of having strong mathematical roots. Several methods rely on auxiliary module [22,23] to compute a confidence score. Most of these algorithms consist in taking the penultimate vector and feeding it both to a classification layer and a confidence one. DeVries et al. [23], for instance, use a confidence score c∈[0,1] to adjust the classifier output $p\in {\left[0,1\right]}^{n}$ with $\sum_{i=1}^{n}p_{i}=1$. The adjustment is as follows: $p_{i}^{\prime }=cp_{i}+\left(1-c\right)y_{i}$ with $y_{i}$ the one-hot encoded ground truth. The method aims to optimize the following loss function: $L=-\sum_{i=1}^{M}log\left(p_{i}^{\prime }\right)y_{i}-\lambda log\left(c\right)$. Finally, Jha et al. [24] propose a non-intrusive approach, based on integrated gradients. Once the model is trained, attributions are computed by dividing the integrated gradient by the associated feature. Then, the attributions are used as a probability to switch the given feature to a baseline. The confidence metric for a sample is given by the resilience of the model across *S* perturbed inputs drawn from this particular sample.

## 3. Materials and Methods

### 3.1. Dataset

The dataset used in this study is from the Brain Tumor Radiogenomic Classification challenge [16], which will be referred to as RSNA-MICCAI data. It includes the following multi-parametric MRI scans: T1w, T1wCE, T2w, and FLAIR. Patients with an MGMT promoter methylation greater than 10% are considered positive or methylated (1); the others are considered negative or unmethylated (0) [16]. From the entire training dataset, 300 samples are methylated, and 274 are unmethylated. These data were split into 70% training data and 30% validation data. MRIs can have three different orientations: coronal, axial, and sagittal. MRI volumes have different depths, heights, and widths. In addition, the competition [16] includes another task: tumor segmentation. Annotation masks comprise the GD-enhancing tumor (ET—label 4), the peritumoral edematous/invaded tissue (ED—label 2), and the necrotic tumor core (NCR—label 1), as described both in the BraTS 2012–2013 TMI paper [25] and in the latest paper to summarize BraTS [16]. The test dataset provided for the challenge is not used here as labels are required for the use of confidence metrics. Instead, as a test set, we use a private dataset comprising 98 patients with glioblastoma and their exact methylation percentages with the T1wCE and FLAIR modalities. Among them, 40 are completely unmethylated (0%), while the 58 others have methylation percentages ranging from 8.4 to 85.4 percent (with a mean of 41.3 and a standard deviation of 21.04). The whole methylation percentage distribution can be found in Appendix A. The MRIs used in the private dataset were performed in two centers: the Toulouse University Hospital and the Claudius Regaud Institute. To perform DNA extraction from formalin-fixed paraffin-embedded (FFPE) tissues, MaxwellR RSC DNA FFPE Kit (Promega, AS1450, Madison, WI, USA) is used. DNA is quantified by fluorimetric assay with DNA Qubit, Broad Range Kit (Thermo Fisher Scientific Q32853, Eugene, OR, USA) and purity is checked by NanoDrop ND-100 (Thermo Fisher Scientific, Eugene, OR, USA). We use pyrosequencing technology (PyroMark Q24, Qiagen, Hilden, Germany) for real-time, sequence-based detection and quantification of MGMT sequence. Prior to pyrosequencing Bisulfite conversion is performed with EZ DNA Methylation Kit (ZYMO RESEARCH, D5001, Irvine, CA, USA) according to the supplier’s recommendations as well as all the steps leading to MGMT pyrosequencing with the MGMT Pyro kit (Qiagen, 970032, Hilden, Germany). MGMT Pyro Kit is used for quantitative measurement of methylation level in region +17 to +39 of exon 1 of the human MGMT gene (ENSG00000170430/ENST00000306010.8)

### 3.2. Data Preprocessing

#### 3.2.1. RSNA-MICCAI Data

We conducted both 3D and 2D experiments; then, we chose to focus on 3D to fully utilize the depth of the tumor. We use the NIfTI files provided with the segmentation task and the DICOM files provided with the radiogenomic task. In each case, the scan resolution is fixed to 180 × 180 × 64, when considering the whole volume, and 96 × 96 × 32, when considering the tumor only. The tumor’s region of interest is extracted thanks to segmentation masks and by considering all of the labels (from the necrotic core to the GD-enhancing tumor). For contrast enhancement, we apply multidimensional contrast limited adaptive histogram equalization, which is a common technique in biomedical image analysis [26]. Finally, each volume *x* is normalized: $x_{scale}=\frac{x-x_{mean}^{*}}{x_{std}^{*}}$ where $x^{*}$ stands for non-zero pixel values. For each experiment, volumes are augmented on the fly with random noise, random blurring, rotations, shifts, and flips.

#### 3.2.2. Private Dataset

In addition to the processing performed on the RSNA-MICCAI data, in accordance with the routine described in the competition paper [16], we perform the following steps: conversion to NIfTI format, co-registration to the same anatomical template (SRI24) [27], resampling to uniform isotopic resolution, and skull-stripping. This preprocessing pipeline is publicly available through the Cancer Imaging Phenomics Toolkit (CaPTk) [28,29,30]. Moreover, to obtain segmentation masks, we train a UNETR [31] model on the BraTS competition data [16] to then extract the segmentation information from the private dataset. We choose to use only one label for ground truth: the fusion of the different classes (necrotic core, peritumoral edematous/invaded tissue, GD-enhancing tumor) to improve the overall performance of the algorithm. For each mask, an anomaly detection algorithm is used to avoid having cubes that are too large around the tumor. The learning curves are available in Appendix A.

### 3.3. Deep Learning Models and Results

Based on the work performed by Saeed [15], the Kaggle competition participants [16], and our preliminary experiments, we make the legitimate assumption that this problem is model-agnostic. Models based on 3D convolution are proven and show impressive results when dealing with medical imaging [32,33,34]. For the sake of clarity and thoroughness, we choose to train 3D convolution models and, in particular, a state-of-the-art ResNet10-3D, which is also the winning architecture of the Kaggle challenge [16]. Finally, data sparsity is a scourge both in our problem and globally in medical image tasks. To tackle this issue and leverage the benefits of pre-trained weights, we sometimes use weights trained on large medical datasets [35]. Each layer of the model is still trainable, none is frozen. This serves as a reference for the confidence results obtained. To take advantage of the multitude of data at our disposal, we conduct experiments both with unimodal and multimodal MRI scans. For multimodal data, we explore 3 different types of fusion.

Early fusion: it refers to the process of joining multiple input modalities into a single feature vector before feeding into one single machine learning model for training.Intermediate fusion: Modalities are joined at the embedding level; we use the EmbraceNet architecture [36]. In comparison to the original paper, we add a layer of multi-head attention [37] between each modality embedding.Late fusion: It refers to the process of leveraging predictions from multiple models to make a final decision.

Table 1 outlines that, on our handmade validation set, we can achieve comparable results to the ones found in [15] and on the Kaggle Competition leaderboard [16].

## 4. Confidence Scores & OOD Detection

The results obtained, in accordance with those of Saeed et al. [15], are rather discouraging and cannot be safely used for a clinical application, contrary to other findings in the literature [10,11,12]. We seek to verify that it is not possible to extract any accurate and useful information. According to the competition paper [16], patients with a methylation percentage above 10% were interpreted as methylated, and the ones below were interpreted as unmethylated. Such a hard threshold raises some concerns: what is, concretely, the anatomic difference between a patient with a methylation percentage of 9.9 and another patient with a percentage of 10.1? At first glance, the radiogenomic characteristics would be the same, and yet one is labeled as unmethylated and the other as methylated. We make the assumption that, in some samples, the distinction between these two classes is not possible, thus resulting in a low general performance for our model.

We focus on rejecting hard inputs (a methylation percentage close to 10%) and evaluating only clean, well-distinguished ones. For example, we hope that a model is better when classifying a patient with a very high methylation percentage, around 60%, or a completely unmethylated one (0%). For this purpose, we use different confidence metrics to assert or refute this hypothesis: the softmax-based one [20], the confidence branch introduced by DeVries et al. [23], the temperature scaling [21], and the ABC metric proposed by Jha et al. [24]. For this last metric, we normalize the probability by patches of the size 6 × 6 × 2 because, when dealing with such high-dimensional inputs, we are directly subjected to the curse of dimensionality with a near-zero probability value for each feature.

For each method presented above, we obtain a confidence distribution D. We look at our model’s performance as the decision threshold is pushed toward $\max_{D}$. Confidence thresholds are set at arbitrary percentiles: 0, 25, 50, 70. Accuracy is calculated as follows: if δ, the model output, is greater than 0.5, the predicted class is set to 1; otherwise, it is set to 0.

Table 2 and Figure 1 outline the poor and unusable results on the validation set despite the proven nature of the techniques employed. The presented results include only early fusion on the FLAIR and T1wCE modalities, but they provide an accurate and meaningful picture of the scores obtained for different configurations. The results obtained with different input configurations are presented in Appendix B.

Private dataset: for this dataset, we have an extra piece of information, the exact methylation percentage. Therefore, we can directly check if a deep learning model performs better on samples spaced from the decision threshold. Thus, an MGMT promoter with a methylation percentage near zero or, inversely, a high methylation percentage, may be easier for our model to identify. To exploit this percentage, we derive an arbitrary surrogate confidence metric to fit the private dataset and our objective. Such a metric is defined in Equation (Equation 1).
(1)$$\begin{array}{c} c\left(\kappa ;x\right)=\left\{\begin{array}{cc} x-\kappa & \mathrm{if}\;x>\kappa \\ \left(\kappa -x\right)e^{\frac{x}{\alpha }} & \mathrm{otherwise} \end{array}\right. \end{array}$$

For this study, we choose κ=10 and α=5.5. Since we have 98 patients, among whom 40 have the same methylation percentage (0%), the subgroups differ from the ones presented in the validation set. Indeed, the metric defined in Equation (Equation 1) only isolates 74, 50, and 46 percent of the test set. The 46 percent subgroup corresponds to all unmethylated patients, while the 50 percent subgroup also includes patients with very high methylation values. In the same way as with our handmade validation set, we can study the ROC and PR curves of the model on the private dataset.

The results presented in Figure 2 also reflect an inability of the model to correctly classify intrinsically better-separated tumor samples. This corroborates with the results obtained on the validation set and with methods from the literature [15].

## 5. Discussion

The MGMT methylation percentage is a very important indicator for neuro-oncologists. This information allows clinicians to personalize and optimize treatment for each patient. Given the many breakthroughs of deep learning in the medical field, one can legitimately try to use such algorithms to predict MGMT methylation status. This question is, therefore, the subject of much research and contradiction. For some, this is completely possible, and their work has yielded results indicating that these models can be used by clinicians in the decision-making process. Nevertheless, a neuro-oncologist interested in such a tool only has at his disposal the Kaggle competition [16] models and the code to train them. The latter have very poor performance and is close to randomness. These different results can be due to several factors. For instance, Chang et al. [12] admit in their discussion that their results must be considered with caution due to the small size of their sample (n=256). Moreover, they point out that according to the TCIA dataset they use [18], their results mean that the convolutional method is capable of dealing with nonuniform imaging protocols. Finally, they mention the lack of an independent test set to completely evaluate their approach. Yogananda et al. [10] achieve impressive results on a 3-folds validation set. We attempted to reproduce their technique, focusing on regions of interest near the tumor to obtain similar results, in vain. This method does not improve model performance. In their study, each fold is considerably small (82, 81, and 82 patients respectively), which increases error margin on evaluation. According to Saeed et al. [15], they and other researchers tried to reproduce the method developed by Yogananda et al. [10] but could not achieve such performance. Furthermore, they were not able to obtain a response from the authors. The code produced to obtain their result is not publicly available and, de facto, not reproducible. Then, the work made by Korfiatis et al. [11] is based on a single acquisition source with a few pre-processing steps that might lead to biased results. Moreover, the method used is a 2D approach, resulting in a classification with many slices without tumor (≈78%). These types of methods are prone to data leakage between the training, validation, and test sets. Similarly, experiments based on the radiomic features extracted from MRI scans yield highly accurate results [13,14]. These results are obtained on a really small cohort (53 patients) that might be the source of a significant error margin. The source code provided with these papers is not directly usable; the paper of Do et al. [13] lacks the feature selection part with XGBoost. Using the parameters given in their work for feature selection, the results obtained differ drastically and are close to random—the same as our results. The section on feature selection to predict MGMT methylation status remains unclear. Le et al. [14] used F-scores, while Do et al. [13] relied on the XGBoost gain score attributed to each feature. It seems that this part is completed before the cross-validation and would, therefore, be adjusted on the whole dataset, representing a huge data leakage and leading to completely erroneous results. When testing the code with the selection before the splitting part, we find the results announced by the authors. On a more pessimistic note, the approach of Saeed et al. [15] is much more systematic and evaluates different state-of-the-art deep learning architectures on this task. This approach leads to less impressive results, unusable by a clinician. The work of Saeed et al. [15], in line with the results of the Kaggle competition [16], corrects this doubt. However, the use of raw performance alone is insufficient for assessing the feasibility of predicting MGMT promoter methylation status. Indeed, the arbitrary separation at 10% between the two classes does not seem particularly suitable to train a deep learning model [16]. In this work, we seek to answer this question once and for all and to deliver fully reproducible results with open-source code. See Appendix C for further details. To achieve this, it is essential to evaluate confidence metrics on this task to see if it is possible to obtain a better performance by focusing on the examples for which the model is the most confident. If the performance on the examples with the highest confidence score is encouraging, this would best assist the clinician in the decision-making process. Indeed, with a high confidence score, the model outputs can be taken into consideration, while, with a low confidence score, the treatment path must be the same as if there were no model at all. We hypothesized that the hard threshold may be the cause of the poor results and the discrepancies in the literature findings and that by considering patients far from the decision frontier, our results would improve. To achieve this, we used proven confidence metrics from the literature on TCGA [19] and TCIA [18] data with no significant results. We also had valuable information from an independent test set: the exact methylation percentage. We were able to derive a confidence metric and observe that the deep learning models do not perform better in the presence of highly methylated or weakly methylated patients than in the presence of patients with methylation percentages near the decision frontier.

## 6. Conclusions

Through this study, we provide a rigorous and systematic answer to the question of predicting the status of the MGMT methylation promoter. Confidence metrics, as well as the exact percentage of methylation, are used to push the reasoning further and decide more precisely. Indeed, determining the data on which deep learning would perform well not only explains the discrepancies in the results in the literature but also provides the clinician with additional information on the value he can choose to attribute, or not, to the output of the deep learning model. The numerous experiments conducted allow us to conclude that it is not possible with the algorithms developed so far to extract information on the MGMT biomarker from anatomical MRI scans.

## Figures and Tables

**Figure 1 cancers-15-02253-f001:**
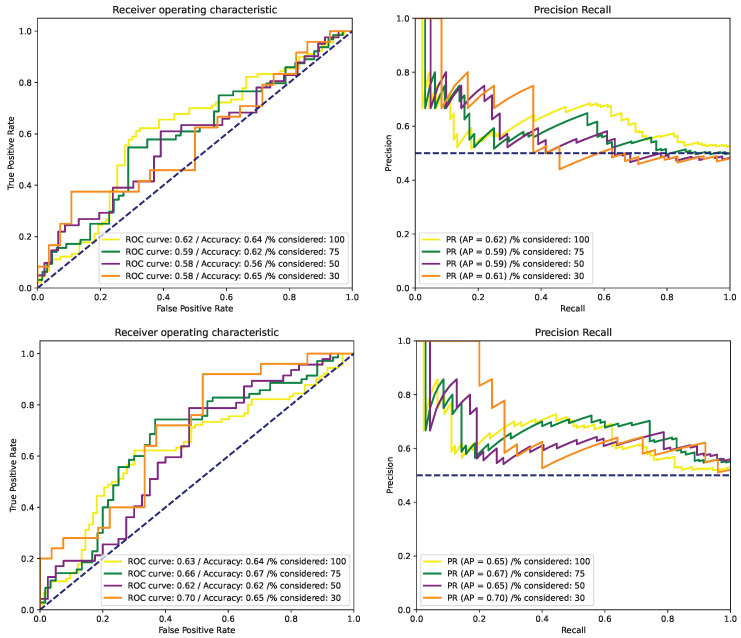
ROC and PR curves with FLAIR-T1wCE inputs for (**top**) baseline [20] and confidence branch [23] (**bottom**).

**Figure 2 cancers-15-02253-f002:**
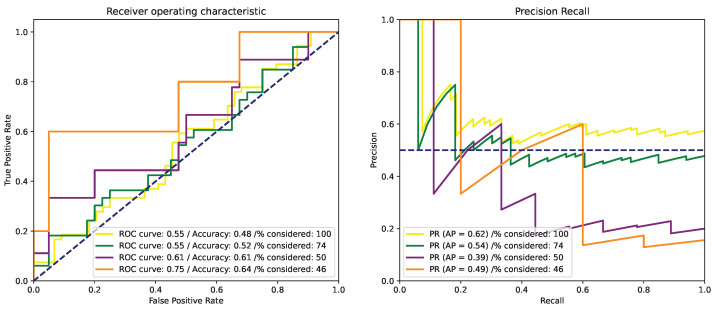
ROC and PR curves for the methylation percentage metric.

**Table 1 cancers-15-02253-t001:** Results on the handmade validation set. Pret. stands for Pre-trained; LR for Learning Rate. When fusion type is set to None, it means that we are using unimodal scan and no fusion is needed.

Pret.	Modality.ies	Fusion Type	LR	Val AUC	Val Acc
Yes	FLAIR	None	1 $\times \;10^{-5}$	0.65	0.62
Yes	T1wCE	None	1 $\times \;10^{-5}$	0.60	0.58
No	FLAIR-T1wCE	early	1 $\times \;10^{-5}$	0.62	0.64
Yes	4 MRI	late	1 $\times \;10^{-3}$	0.63	0.64
Yes	FLAIR-T1wCE	intermediate	1 $\times \;10^{-5}$	0.63	0.61
No	4 MRI	early	1 $\times \;10^{-5}$	0.60	0.59

**Table 2 cancers-15-02253-t002:** Results of confidence metrics on the validation set. FLAIR-T1wCE, early fusion. A value set to - reflects the fact that the technique was unable to provide the necessary confidence distribution for the split on the desired percentiles. In these cases, the results displayed are those of the possible splits.

Method	% Considered	Val AUC	Val AP	Val Acc
Baseline	100	0.62	0.62	0.64
	75	0.59	0.59	0.62
	50	0.58	0.59	0.56
	30	0.58	0.61	0.65
ODIN	100	0.62	0.62	0.64
	75	0.59	0.59	0.62
	50	0.58	0.59	0.55
	30	0.60	0.63	0.65
Confidence Branch	100	0.63	0.65	0.64
	75	0.66	0.67	0.67
	50	0.62	0.65	0.62
	30	0.70	0.70	0.65
ABC Metric	100	0.62	0.62	0.64
	-	-	-	-
	55	0.49	0.44	0.60
	-	-	-	-

## Data Availability

The data presented in this study are from the https://www.kaggle.com/competitions/rsna-miccai-brain-tumor-radiogenomic-classification/data (last acccessed on 28 March 2023) The code is freely available at https://github.com/Lucas-rbnt/MGMT_conf (last acccessed on 28 March 2023).

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
