# Peer review of "MRI-Based Deep Learning Tools for MGMT Promoter Methylation Detection: A Thorough Evaluation"

_cancers, 2023, doi:10.3390/cancers15082253_

Round 1

Reviewer 1 Report

The article touches on an important aspect of work with great clinical implementation. Good job!

The authors are using a dataset of MRI images using machine learning to identify the level of MGMT promoter methylation status in order to determine possible responses to treatment. The authors conclude that this is not possible.

As a practicing neurooncologist, I obviously find it relevant and original - but not groundbreaking. The idea the Authors propose is a dream, that we will not reach in the near future.

I think negative results are very important. Publishing negative results helps the field not to push areas or research unnecessarily. The study uses deep learning to study radiography images to find out a biological feature.  I don’t think it is possible. I think the methodology is decent.

The conclusions are consistent with the evidence and arguments presented and address the main question posed. The references are appropriate.

Author Response

The authors would like to thank the reviewer for his time and valuable feedback. All changes to the manuscript are highlighted in red in the revised version.

We hope that the changes in the introduction address the reviewer's comments.

Reviewer 2 Report

Beyond the classical performance figures usually displayed, the proposed method is to compute confidence scores to 11 see if a clinical application of such methods can be seriously considered. 

The comments are as follows:

- You should add the comparison of various existing experimental results.

- Various evaluations of the tool should be made as results may vary for different datasets.

- If it is an evaluation of tool use, the consequential aspect of the experiment is too insufficient.

Author Response

The authors would like to thank the reviewer for his time and valuable feedback. All changes to the manuscript are highlighted in red in the revised version.

1. You should add the comparison of various existing experimental results.

Considering your point, we have, in the revised version of the manuscript, further developed the introduction on these methods to make more sense and link with the following parts.
Moreover, we discussed performances in the Related Work and the Discussion sections.
In the Discussion, we provide some answers as to why such results diverge between the literature and our experiments.

2. Various evaluations of the tool should be made as results may vary for different datasets.

As with the previous point, we have taken your comment into account to explain our approach and our reasoning more clearly in the Introduction.
Moreover, the MICCAI-BraTS competition already brings together several datasets to form the largest multimodal cohort on the subject of MGMT promoter methylation classification [1]. In our study, we took the reasoning further by using confidence metrics on one hand, and the exact methylation percentage on a private dataset from The Toulouse University Hospital and the Claudius Regaud Institute on the other.

3. If it is an evaluation of tool use, the consequential aspect of the experiment is too insufficient.

In our study, the contribution we wish to make is fourfold:
(a) Provide a rigorous and complete answer on the feasibility of predicting MGMT promoter methylation where research is still ongoing by implementing state-of-the-art confidence metrics, a key issue to apply deep learning for medical decision-making.

(b) Using the prism of confidence metrics, we seek to explain the discrepancies in results between the literature and the Kaggle competition and to have a more robust and above all definitive answer as to the potential use of Deep Learning algorithms by clinicians on the determination of MGMT promoter methylation status.

(c) Until now, only binary methylated, unmethylated information was taken into account, which is not necessarily relevant to study the feasibility.
In our paper we took into account the exact methylation percentage under the assumption that it could explain the differences in results and would allow us to isolate the cases for which our algorithms would perform well, in vain.

(d) To the best of our knowledge, we provide the first clear source code with fully reproducible experiments, from model training to confidence metric computations and private data management with exact methylation percentage. The code is available here: https://github.com/Lucas- rbnt/MGMT_conf

We hope that theses points and the changes made in the introduction and the conclusion meet your expectations concerning the background and references provided in the introduction and that our conclusions are supported by the results.

Thank you again for your feedback.

References:
[1] Baid, U.; Ghodasara, S.; Mohan, S.; Bilello, M.; Calabrese, E.; Colak, E.; Farahani, K.; al. The RSNA- ASNR-MICCAI BraTS 2021 Benchmark on Brain Tumor Segmentation and Radiogenomic Classification. Technical report, 2021. Publication Title: arXiv e-prints ADS Bibcode: 2021arXiv210702314B Type: article.

Reviewer 3 Report

In this study, the authors try to compute confidence scores to test the feasibility of deep learning methods in the application of MGMT. The results revealed that current deep learning tools are not able to determine MGMT promoter methylation from MRI data. Overall, I think the innovativeness of the article is insufficient. My main question is since there are already relevant papers that have conducted similar research, Is it necessary to add a new quantitative method (confidence scores) to further prove? Furthermore, it is well known that there are no deep learning methods that can 100% accurately classify disease and normal people. Several other questions are as follows,

  1. In the abstract, I didn’t see any association between the biomarker MGMT and the multimodal MRI scan. Why can multimodal MRI obtain this information? Hope can add some interpretation. Furthermore, there are too many introductions in the clinical part, which can be shortened appropriately.
  2. Similarly, in the first paragraph of the introduction, please introduce why multimodal MRI can extract MGMT status information.
  3. I hope the author can give some specific parameters, such as under what circumstances are deep learning based classification methods clinically acceptable? 100%? 90%? I think deep learning methods are not able to determine MGMT promoter methylation from MRI data is a boring conclusion.

Some minor questions

  1. In line 46, the expression "Then, we reproduce classical results" is vague." what is  "classical results" come from?
  2. In Table 1, I suggest adding an interpretation for "None".
  3. In Table 2, I suggest adding an interpretation for "-" in ABC Metric.

Author Response

The authors would like to thank the reviewer for his time and valuable feedback. All changes to the manuscript are highlighted in red in the revised version.

1. My main question is since there are already relevant papers that have conducted similar research, Is it necessary to add a new quantitative method (confidence scores) to further prove?

Considering your remark, we modify the abstract and the introduction in the revised manuscript to explain our reasoning and approach. Moreover, the interest in using confidence metrics here is twofold. Indeed, the biases and errors of deep learning models when faced with attacks, shifts in the distribution, on "in-between" examples or even on data in the distribution are an obstacle to their use in a critical context. The difference between human errors and those of these algorithms is that for humans, errors can be explained and result from reasoning, whereas deep learning is still considered a black box from which we only look at the prediction. In the medium term, to use deep learning in the medical field, or any other critical system, each prediction must be accompanied by its confidence score. Thus, when making a diagnosis, the clinician could look at the model's prediction and, depending on the confidence score and his or her own expertise, assign a variable importance to it.

In this context, to come back to the question, yes, the use of these metrics seems to us to be essential for medical issues and not to consider it would even be dangerous.
Secondly, in the case of determining the status of the MGMT promoter, the research papers show very promising results in contrast to the results of the Kaggle competition on the same theme. Here the confidence metrics were used to try to explain these differences. Indeed, we worked on the largest public MICCAI-BraTS dataset [1] as well as a large private dataset obtained from The Toulouse University Hospital and the Claudius Regaud Institute (n=98).

The confidence techniques employed, if they had isolated a subset of samples on which the model performed encouragingly, would have explained this difference in results.

2. Furthermore, it is well known that there are no deep learning methods that can 100% accurately classify disease and normal people.

This point is perfectly correct, but one never asks a deep learning model to perform at 100%. Even on the MNIST benchmark (a large database of handwritten digits in grayscale 28x28) deep learning models do not classify accurately 100% of the samples. Many deep learning models are used today in a variety of sectors and are the gold standard for these tasks, yet they continue to make mistakes. In the medical field too, deep learning is often used to assist in diagnosis, image registration, segmentation, classification, drug discovery and so on.

3. In the abstract, I didn’t see any association between the biomarker MGMT and the multimodal MRI scan. Why can multimodal MRI obtain this information? Hope can add some interpretation. [...] Similarly, in the first paragraph of the introduction, please introduce why multimodal MRI can extract MGMT status information.

We have significantly modified the abstract and the introduction to meet your expectations. The new version is more clear, sourced and straightforward. To resume and respond to your review: in our work we considered only the T1w, T2w, T1wCE and FLAIR anatomical modalities. Many studies suggest that it is possible to derive MGMT promoter status based on these MRIs.

For example, Drabycz et al. [2] conducted a retrospective analysis of tumor texture from T2w MRI. They considered on the one hand expert descriptions of tumor edges, pattern of enhancement and on the other hand a space-frequency texture analysis. They also used information on the location of the tumor through automatic segmentation tools. They concluded that there was an association between visually assessed ring enhancement and unmethylated MGMT patients and that there were significant differences between methylated and unmethylated samples on the texture features obtained from space-frequency analysis. Furthermore, Eoli et al. [3] support the finding on the association between ring enhancement and MGMT unmethylated patients. Han et al. [4] considered MR features derived by two neuroradiologists to predict the MGMT promoter status with statisticaltechniques such as Fisher’s exact test and logistic regression. They find out that MGMT promoter methylation was associated with tumor location and necrosis.

However, again, there is no clear consensus with studies showing no association. For instance, Mikkelsen et al. [5] outlined that there is no significant associations between the promoter MGMT status and images features (necrotic volumes, percentage of necrosis, tumor volumes..) derived from T1wCE imagery.

Finally, from a pragmatic standpoint, research papers using machine learning to determine MGMT status with excellent results [6, 7, 8, 9, 10] (sometimes over 90% [6, 9]) attest to the presence of this information directly in the data. Indeed, studies using radiomics-features extracted from MRI scans to build powerful, non-linear models are creating a function between these features (size and shape- based, image intensity histogram...) and the MGMT prediction. Moreover, when training a highly- parametrized deep learning model directly on MRI scans, we are building a mapping function between voxel intensity and the prediction of MGMT promoter status.

4. Furthermore, there are too many introductions in the clinical part, which can be shortened appropriately.

In accordance with your review, we have shortened the unnecessary parts of the introduction to make it more readable and straightforward.

5. I hope the author can give some specific parameters, such as under what circumstances are deep learning based classification methods clinically acceptable? 100%? 90%?

The minimum performance for a deep learning model to be clinically acceptable is relative. In general, giving a performance threshold only makes sense in the case of a common dataset with good performance that one would seek to improve with new paradigms or techniques (e.g., Brain Tumor Segmentation [1]) which is not the case here since we are discussing the task feasibility and the reasons for the differences found in the literature.

Furthermore, there are many cases of classification where AI provides similar or better results than clinicians (BioMind AI for instance). Also, many tools based on deep learning are also used by doctors as a decision-making support (for skin cancer detection for instance).

In the case of MGMT promoter status prediction, the objective was to explain the disparities between the literature results and the MICCAI-BraTS competition [1] as well as some research papers too [11, 5]. This analysis is conducted to decide as scientifically and rigorously as possible on the feasibility of the task, and in the positive case, to create a decision support tool for the clinician, who could refer to. Thus, if it was possible to isolate examples for which deep learning would perform similarly to what can be found in the literature, the clinician could have the prediction of the model as well as the confidence of the model in its prediction.

Both pieces of information could help, or not, the clinician in his work.

6. I think deep learning methods are not able to determine MGMT promoter methylation from MRI data is a boring conclusion. 

Even if this conclusion seems boring and disappointing, it appears essential to us to try to explain the differences between the very good or even excellent results that can be found in the literature and the lack of concrete clinical application of these methods as well as the results of the last MICCAI BraTS competition [1]. We have sought to do this by using the hitherto unused notion of confidence and by working on the exact percentage of methylation, which was also not considered in the previous work and in the competition but which is nevertheless essential. Furthermore, to our knowledge, we provide the first accessible source code with the results of our paper completely reproducible. The code allows one to train a model, to calculate state-of-the-art confidence metrics and to test them on one’s own dataset.

The code is available here: https://github.com/Lucas-rbnt/MGMT_conf

7. In line 46, the expression "Then, we reproduce classical results" is vague." what is "classical results" come from?

We have deleted this unnecessary part of the introduction in accordance with your previous remarks. This involved the results of the Kaggle competition and the research paper by Saeed et al. [11]. The turn of phrase was wrong, sorry.

8. In Table 1, I suggest adding an interpretation for « None »

This has been corrected in the revised manuscript. When fusion type is set to ‘None’, it means that we are using unimodal scan and de facto, no modality fusion is needed.

9. In Table 2, I suggest adding an interpretation for "-" in ABC Metric.

This has been corrected in the revised manuscript. A value set to ‘-’ reflects the fact that the technique was not able to provide the necessary confidence distribution for the split on the desired percentiles. In these cases, the results displayed are those of the possible splits.

Once again, the authors wish to thank the reviewer for his relevant feedback and hope that the improvements to the manuscript made in function as well as the answers help to clarify certain points.

References:
[1] Baid, U.; Ghodasara, S.; Mohan, S.; Bilello, M.; Calabrese, E.; Colak, E.; Farahani, K.; al. The RSNA- ASNR-MICCAI BraTS 2021 Benchmark on Brain Tumor Segmentation and Radiogenomic Classification. Technical report, 2021. Publication Title: arXiv e-prints ADS Bibcode: 2021arXiv210702314B Type: article.

[2] Drabycz, S.; Roldán, G.; Robles, P.d.; Adler, D.; McIntyre, J.B.; Magliocco, A.M.; Cairncross, J.G.; Mitchell, J.R. An analysis of image texture, tumor location, and MGMT promoter methylation in glioblastoma using magnetic resonance imaging. NeuroImage 2010

[3] Eoli, M.; Menghi, F.; Bruzzone, M.G.; De Simone, T.; Valletta, L.; Pollo, B.; Bissola, L.; Silvani, A.; Bianchessi, D.; D’Incerti, L.; et al. Methylation of O 6-Methylguanine DNA Methyltransferase and Loss of Heterozygosity on 19q and/or 17p Are Overlapping Features of Secondary Glioblastomas with Prolonged Survival. Clinical Cancer Research 2007

[4] Han, Y.; Yan, L.F.; Wang, X.B.; Sun, Y.Z.; Zhang, X.; Liu, Z.C.; Nan, H.Y.; Hu, Y.C.; Yang, Y.; Zhang, J.; et al. Structural and advanced imaging in predicting MGMT promoter methylation of primary glioblastoma: a region of interest based analysis. BMC Cancer 2018,

[5] Mikkelsen, V.E.; Dai, H.Y.; Stensjøen, A.L.; Berntsen, E.M.; Salvesen, ; Solheim, O.; Torp, S.H. MGMT Promoter Methylation Status Is Not Related to Histological or Radiological Features in IDH Wild-type Glioblastomas. Journal of Neuropathology and Experimental Neurology 2020.

[6] Yogananda, C.G.B.; Shah, B.R.; Nalawade, S.S.; Murugesan, G.K.; Yu, F.F.; Pinho, M.C.; Wagner, B.C.; Mickey, B.; Patel, T.R.; Fei, B.; et al. MRI-Based Deep-Learning Method for Determining Glioma MGMT Promoter Methylation Status. AJNR. American journal of neuroradiology 2021.

[7] Korfiatis, P.; Kline, T.L.; Lachance, D.H.; Parney, I.F.; Buckner, J.C.; Erickson, B.J. Residual Deep Convolutional Neural Network Predicts MGMT Methylation Status. Journal of Digital Imaging, 2017

[8] Chang, P.; Grinband, J.; Weinberg, B.D.; Bardis, M.; Khy, M.; Cadena, G.; Su, M.Y.; Cha, S.; Filippi, C.G.; Bota, D.; et al. Deep-Learning Convolutional Neural Networks Accurately Classify Genetic Mutations in Gliomas. AJNR. American journal of neuroradiology 2018

[9] Do, D.T.; Yang, M.R.; Lam, L.H.T.; Le, N.Q.K.; Wu, Y.W. Improving MGMT methylation status prediction of glioblastoma through optimizing radiomics features using genetic algorithm-based machine learning approach. Scientific Reports 2022

[10] Le, N.Q.K.; Do, D.T.; Chiu, F.Y.; Yapp, E.K.Y.; Yeh, H.Y.; Chen, C.Y. XGBoost Improves Classification of MGMT Promoter Methylation Status in IDH1 Wildtype Glioblastoma. Journal of Personalized Medicine 2020,

[11] Saeed, N.; Hardan, S.; Abutalip, K.; Yaqub, M. Is it Possible to Predict MGMT Promoter 385 Methylation from Brain Tumor MRI Scans using Deep Learning Models?, MIDL, 2022

Round 2

Reviewer 3 Report

There is nothing wrong with the content of the article, and the innovation of the article has been updated in the introduction. But I still don't think the innovation is significant.